# Potential Antitumor Effect of α-Mangostin against Rat Mammary Gland Tumors Induced by LA7 Cells

**DOI:** 10.3390/ijms241210283

**Published:** 2023-06-17

**Authors:** Mohamed Yousif Ibrahim, Najihah Mohd Hashim, Fatima Abdelmutaal Ahmed Omer, Muhammad Salisu Abubakar, Hoyam Adam Mohammed, Suzy Munir Salama, Soher Nagi Jayash

**Affiliations:** 1Faculty of Pharmacy, Elrazi University, Khartoum 11115, Sudan; 2Department of Pharmaceutical Chemistry, Faculty of Pharmacy, Universiti of Malaya, Kuala Lumpur 50603, Malaysia; najihahmh@um.edu.my; 3Center for Natural Products and Drug Discovery (CENAR), University of Malaya, Kuala Lumpur 50603, Malaysia; 4Department of Chemistry and Biology, Faculty of Education-Hantoub, University of Gezira, Wad Madani 21111, Sudan; fatimaomer51@yahoo.com; 5Department of Veterinary Pathology, Faculty of Veterinary Medicine, Usmanu Danfodiyo University, Sokoto 840212, Nigeria; salisu.abubakar@udusok.edu.ng; 6Faculty of Pharmacy, University of Sinnar, Sinja 25511, Sudan; hemaadam358@gmail.com; 7Indigenous Knowledge and Heritage Center, Ghibaish College of Science and Technology, Sinja 25511, Sudan; s.salama999@hotmail.com; 8School of Dentistry, University of Birmingham, 5 Mill Pool Way, Edgbaston, Birmingham B5 7EG, UK

**Keywords:** α-mangostin, mammary cancer, LA7 cells, apoptosis, immunohistochemistry, antioxidant

## Abstract

In this study, the chemotherapeutic effect of α-mangostin (AM) was assessed in rats injected with LA7 cells. Rats received AM orally at 30 and 60 mg/kg twice a week for 4 weeks. Cancer biomarkers such as CEA and CA 15-3 were significantly lower in AM-treated rats. Histopathological evaluations showed that AM protects the rat mammary gland from the carcinogenic effects of LA7 cells. Interestingly, AM decreased lipid peroxidation and increased antioxidant enzymes when compared to the control. Immunohistochemistry results of the untreated rats showed abundant PCNA and fewer p53-positive cells than AM-treated rats. Using the TUNEL test, AM-treated animals had higher apoptotic cell numbers than those untreated. This report revealed that that AM lessened oxidative stress, suppressed proliferation, and minimized LA7-induced mammary carcinogenesis. Therefore, the current study suggests that AM has significant potential for breast cancer treatment.

## 1. Introduction

Breast cancer is a malignancy that poses a significant threat to women globally, owing to its elevated incidence and mortality rates [1]. As per the American Cancer Society, breast cancer constitutes the largest proportion of newly diagnosed cancer cases among American women. It accounts for 26% of all such cases. According to the National Cancer Registry of Malaysia, the lifetime risk of breast cancer for women in Malaysia is estimated to be one in twenty [2]. The current incidence rate, compared to one in eight women in Europe and the United States, is relatively low. Environmental factors and lifestyle are reported to cause up to 70% of breast cancers in women. The remaining 30% can be attributed to genetic components [3].

Reactive oxygen species (ROS) appear to play a significant role in various critical pathophysiological conditions, such as mutagenesis and carcinogenesis [4]. Free radicals are significant in tumor promotion, as they can directly react chemically or modify cellular metabolic processes. Free radical scavengers, such as SOD, CAT, and GPx, act as inhibitors at various stages of carcinogenesis. These enzymes are present in both cytosolic and mitochondrial functions and are primarily involved in the biotransformation and detoxification of carcinogens [5]. The ongoing severity and magnitude of cancer problems necessitate the development of chemotherapy strategies based on natural antioxidants. This is to block the initiation or arrest/reverse the progression of premalignant cells [6]. Antioxidants protect against reactive oxygen species (ROS) due to their ability to impede ROS formation [7]. Antioxidant concentrations are crucial to counteracting oxygen-free radicals [8]. Antioxidants have been associated with life-threatening illnesses such as cancer [9]. 

The development of a novel strategy with anti-neoplastic and free-radical-scavenging properties is crucial for various reasons. Hence, it is pertinent to explore diverse phytotherapeutic sources for anti-neoplastic and antioxidant properties [10]. Currently available synthetic anticancer medications, such as Tamoxifen, serve as the drug of choice for estrogen-dependent breast cancer. Tamoxifen’s antitumor effect is attributed to stopping the growth of the tumor and apoptosis being triggered. This occurs by binding to the intracellular estrogen receptors in breast cancer cells. It also blocks steroid hormones and stops protein kinase C from binding to calmodulin. However, there are various side effects such as liver cancer, increased blood clotting, retinopathy, and corneal opacities when performing chemotherapy with Tamoxifen [11]. Due to these side effects, researchers are investigating the potential of finding potent phytotherapeutic agents with non-cytotoxic characteristics [12].

In this regard, xanthones are naturally occurring chemical compounds present in various fruits and vegetables. Additionally, these compounds exhibit diverse biological, biochemical, and pharmacological properties. This indicates their significant impact on fundamental cellular processes such as proliferation, differentiation, and/or apoptotic pathways [13]. Among the available xanthones, recently, mangostin compounds have gained much attention due to their efficacy in fighting diseases, especially cancer and cancer-related diseases. Among the mangostins, α-mangostin (AM) (Figure 1) has a wide spectrum of biological activities such as antioxidant, anti-inflammatory, cardio-protective, antitumor, anti-diabetic, antibacterial, antifungal, and anti-parasitic properties, and it acts as an anti-obesity agent [14].

Animal models are usually utilized to evaluate the chemotherapeutic effects of novel drugs from different origins. LA7-induced mammary carcinogenesis is one of the recent mammary cancer models and is the most similar to the human breast cancer form [15,16]. This study was conducted to determine the chemotherapeutic effect of AM isolated from mangostin fruit on LA7-induced mammary tumorigenesis.

## 2. Results

### 2.1. Tumor Development

Mammary tumors were observed 7 to 10 days after LA7 cell injections, and they indicated strong tumorigenic properties. The tumors were soft and rubbery, and as they grew, they became irregular and lobulated.

### 2.2. Body Weight, Tumor Size, and Tumor Volume

Table 1 depicts the body weight, tumor volume, and tumor percentage reduction (%) of untreated and treated rats. The body weight was significantly (*p* < 0.05) reduced in the mammary tumor control (MTC) group of rats when compared with normal control (NC) rats. In contrast, the groups administered with AM 30 mg/kg (low dose -LD), AM 60 mg/kg (high dose -HD), and Tamoxifen (TAM) exhibited significant (*p* < 0.05) increments in their body weights when compared to the MTC group of animals. The MTC group tumors grew rapidly, reaching an average volume of 423 ± 71.2 mm^3^ by day 28. Meanwhile, groups treated with AM-LD and AM-HD showed a significant (*p* < 0.05) reduction in tumor size in comparison with the MTC group. The AM-HD also resulted in further reduction (71.4%) in the tumor size in comparison to the size (56.3%) obtained from the AM-LD. The effect of AM 30 mg/kg, AM 60 mg/kg, and TAM on animal tumor volume (mm^3^) compared to the control groups throughout the experimental period is presented in Figure 2.

### 2.3. Blood Biochemical Parameters

In this study, blood samples were obtained from rats at the end of the treatment period and used to determine biochemical parameters. Table 2 illustrates the effect of AM on the levels of the following tumor markers in the serum of control and experimental animal groups: carcinoembryonic antigen (CEA), breast-cancer-specific marker (CA15-3), alanine aminotransferase (ALT), alkaline phosphatase (ALP), and lactate dehydrogenase (LDH). In MTC-group rats, there was a significant (*p* < 0.05) rise in the measurements of these tumor markers compared to normal control rats (NC). On the other hand, measurements of these tumor markers were markedly (*p* < 0.05) decreased with AM treatment (MT + AM-LD and MT + AM-HD) in comparison to the MTC animal group, which was possibly due to a decrease in tumor development, revealing AM’s therapeutic effect. In addition to that, there was a significant increase (*p* < 0.05) in ALT concentrations in the experimental groups (II–V) compared to the normal control group of animals (NC). However, the level of ALT significantly (*p* < 0.05) decreased upon AM treatment (MT + AM-LD and MT + AM-HD) in comparison with the MTC group of animals. Moreover, the serum ALP concentration was significantly (*p* < 0.05) higher in experimental groups II–V than in controls. Further, the concentration of ALP was considerably (*p* < 0.05) reduced with AM treatment (MT + AM-LD, MT + AM-HD) and MT + TAM when compared to the MTC group of animals. In our study, the serum LDH concentrations decreased (*p* < 0.05) remarkably in all of the treated groups (III–V) when compared to the MTC group.

### 2.4. Histopathology

LA7-induced mammary gland tumors in rats were scored according to tubular formation, nucleus pleomorphism, and mitotic count (Table 3). According to histopathological results, the tumors were of the non-tubular subtype of invasive adenocarcinoma. The tumors also displayed marked nuclear variation in size and shape, a high mitotic index, and necrosis consistent with adenocarcinomas (Table 3 and Figure 3). AM 30 mg/kg b.w., AM 60 mg/kg b.w., and TAM 10 mg/kg b.w. significantly reduced mitotic events and facilitated tissue reorganization (*p* < 0.05) when compared to untreated rats. However, AM-treated groups displayed intra-tumor vascularization and tumor tubules. Normal rats treated with AM demonstrated normal mammary gland morphology. The results further showed improved mammary gland structures following AM treatment compared to untreated rats.

### 2.5. Apoptosis

Mammary gland sections of rats injected with LA7-induced mammary carcinoma after treatment with AM 30 mg/kg body weight, AM 60 mg/kg body weight, and TAM 10 mg/kg body weight showed significantly (*p* < 0.05) higher records of apoptotic cells than the normal control and mammary tumor control rats. In fact, the control rats showed either very little or no apparent apoptosis in the mammary glands (Figure 4A). Apoptotic cells were evident in AM-treated rats (Figure 4C,D).

### 2.6. Antioxidant Activity

In vivo antioxidant evidence of AM and alterations in its free radical scavenger enzymes were determined in the liver supernatants (Table 4). Outcomes showed significantly raised levels of superoxide dismutase (SOD) in AM-LD- and AM-HD-treated groups versus SOD levels in the MTC group of rats. The SOD level was higher in AM-HD-treated rats versus the AM-LD-treated group. Catalase (CAT) activity was significantly (*p* < 0.05) lower in the AM-LD-treated group than the normal control group (NC). Moreover, higher levels of CAT activity were recorded in the group exposed to TAM 10 mg/kg compared to other groups. Meanwhile, the cancer control group (MTC) exhibited less activity and CAT formation than the other groups. Glutathione peroxidase (GPx) measurements were significantly (*p* < 0.05) elevated in the AM 60 mg/kg treated group compared to the group of rats exposed to AM 30 mg/kg. In the cancer control group (MTC), the measurement of GPx was lower than in the normal group (NS). Breast-cancer-bearing rats (MTC) showed higher measurements of lipid peroxide, while AM-LD rats showed a slight decrease. However, the potential reduction of lipid peroxide (LPx) was recorded in the AM-HD and TAM (10 mg/kg) dosed groups, and it was almost equal to the normal control.

Concurrently, the free radical scavengers, superoxide dismutase (SOD), catalase (CAT), glutathione peroxidase (GPx), and lipid peroxide (LPx) measurements in the breast supernatants were recorded, as shown in Table 5. The activities of SOD and CAT were much higher in the mammary glands of cancer-bearing animals dosed with AM 60 mg/kg than in AM 30 mg/kg treated rats. However, CAT activity was elevated (*p* < 0.05) in the AM 60 mg/kg dosed group versus the cancer control rats (MTC). This raised measurement was near that of the normal control rats compared with other groups. The GPx levels were higher in AM (LD and HD)-treated groups as well as TAM (10 mg/kg) treated groups versus cancer control rats (MTC). LPx levels were significantly elevated in breast-cancer-bearing animals, whereas LPx levels were significantly reduced in AM (LD and HD)-treated groups.

### 2.7. Immunohistochemistry of PCNA and p53

The positive effect of AM treatment on mammary gland tumors in rats was confirmed via immunohistochemistry images of proliferating cell nuclear antigen (PCNA) and p53, as shown in Figure 5 and Figure 6. Results indicated that α-mangostin inhibited PCNA expression and augmented p53 expression in the mammary gland tumors in a dose-dependent manner. Treating the tumors with AM-HD and AM-LD revealed significant down-regulation of PCNA in comparison to the negative control group MTC, while the data obtained from AM-HD-treated rats were very close to that obtained from those treated with the standard drug TAM (10 mg/kg). On the other hand, administration of α-mangostin to rats’ tumors showed significant up-regulation of p53 in AM-HD- and AM-LD-treated rats compared to the negative control group MTC. In addition, the record data from the AM-HD group (60 mg/kg) approached that from the standard drug TAM (10 mg/kg). The immunohistochemistry results from the posted images were confirmed by the % of immunopositive cells calculated for both markers tested, PCNA and p53, as illustrated in Table 6.

## 3. Discussion

This study set out to assess the importance of natural compounds as chemotherapeutic agents. Although there have been different methods for tumor induction, such as chemical carcinogens [17] and ionizing radiation [18], they are time-consuming and hazardous. In this study, we used the rat mammary tumor cell LA7 to induce mammary tumors on the left or right flanks of rats. This was intended to produce malignant tumors. Using this method, the mammary gland tumor developed at the injection site within 7 to 10 days. The advantages of this tumor induction method include ease of inoculation method, continuity and reproducibility of tumor growth, safety, and economic benefits. In addition, the tumor rarely expresses cell surface molecules different from those in the tissue of origin [15,16].

Evidence indicates that α-mangosteen (AM) inhibits various cancer cell lines [14,19,20]. The present study shows that the rate of tumorigenicity of LA7 cells in these rats was 80%, the body weight was decreased, and the tumor volume was statistically increased (*p* < 0.05). On the other hand, the AM (60 mg/kg) treated group showed strong chemotherapeutic activity, a decreased tumor volume by an average of 71.4%, and a slight increase in body weight after treatment. In addition to that, treatment with AM (30 mg/kg) significantly diminished the tumor volume by an average of 56.3%, whereas in TAM-treated groups (10 mg/kg) the mammary tumors were reduced by an average of 78.7%. However, there was a loss in body weight observed in all of the groups compared to the normal control rats. Our AM results corroborate other reported antitumor activities [21,22].

Serum biomarkers for breast cancer have been extensively studied for both diagnosis and prognosis of the disease. Many of these biomarkers, including mucins, oncofetal proteins such as carcinoembryonic antigen (CEA), oncoproteins such as receptor tyrosine-protein kinase cellular tumor antigen (p53), and cytokeratins such as tissue polypeptide antigen (TPA) and tissue polypeptide specific antigen (TPS), have been recommended for breast cancer. The most common and clinically detected markers are carcinoembryonic antigen (CEA) and carcinoma antigen 15-3 (CA 15-3). Since the 1960s, CEA has been used as a biomarker. It is a glycoprotein found in the embryonic endodermal epithelium that is up-regulated in cancerous epithelial-type tumor cells. Serum CEA concentrations are higher in colorectal, breast, lung, liver, and pancreatic cancers [23,24]. This increase in serum CEA in the MTC group of animals is one piece of evidence that the rats injected with LA7 developed mammary gland cancer.

CA 15-3 is a transmembrane glycoprotein of a serum-based derivative of the MUC1 gene and is the most extensively utilized biomarker in observing progressive breast cancer cases that overproduce CA 15-3 in response to chemotherapy [25], and it is commonly expressed through breast cells. In malignant mammary gland tumors, there is an augmented production of CA 15-3. It is absorbed by the malignant cells and moves into the blood circulation, making it valuable as a tumor marker to follow the course of chemotherapy in order to cure the cancer [25]. Concomitant with the above, tumor markers were significantly reduced with both AM doses tested. This indicated that rats treated with AM had a favorable prognosis with chemotherapy.

Liver serum enzymes ALP, ALT, and LDH are most useful for monitoring therapy response and detecting toxicity [26]. In one study, a significant increase in ALP was observed in the bone and liver metastases of breast cancers [26]. During the treatment period, although the serum ALP concentration decreased, it still remained significantly higher than in the controls. This study suggests that AM-LD, AM-HD, and TAM treatment significantly affect serum ALP concentration in tumor-bearing rats, concurring with a previous report regarding the evaluation of acute and sub-chronic AM toxicity [27]. It should be noted that ALP is not an organ-specific enzyme, as it is present in many tissues of the body. However, ALP examination together with CA 15-3 detection can be useful in mammary gland tumor diagnosis [28]. Liver disorders can be determined from serological markers such as alanine transaminase. The abnormal increase in serum ALT concentrations may suggest liver metastasis [29] and hepatotoxicity [30]. At the end of the study, TAM-treated rats showed a significant increase (*p* < 0.05) in serum ALT. It is reported that TAM treatment can significantly boost ALT serum activity, which is correlated with hepatotoxicity [31]. The serum LDH concentration significantly increases in patients with endometrial adenocarcinomas, ovarian adenocarcinomas, and breast cancers [32]. In our study, the rats treated with AM-LD, AM-HD, and TAM showed a decrease in serum LDH that reflected lower tumor volume, resulting in decreased release of cytoplasmic LDH when compared to the mammary tumor control group of animals. Therefore, AM may represent an effective chemotherapeutic drug for cancer treatment. These results are consistent with those of a previous case report study [33].

It is essential for the body to maintain an active equilibrium between the number of free radicals produced and the antioxidant defense system. This eliminates them and protects the body from pathogenic effects [34]. Free radicals exist both at the beginning and the end of carcinogenesis, and their biochemical responses in each phase of the metabolic process are correlated with cancer progression [8]. It is evident from the results that augmented concentrations of LPx were observed in the mammary cancer-bearing group (MTC) when compared to the control group. On the contrary, reduced levels of LPx were observed in the AM (LD and HD)-treated groups indicating that it is an effective free radical scavenger.

Endogenous antioxidant enzymes perform as the crucial line of protection against reactive oxygen species (ROS), and this suggests their effectiveness in assessing the risk of oxidative harm induced throughout carcinogenesis. SOD is the major antioxidant enzyme that deals with oxy-radicals by quickening the partition of superoxide (O^2−)^ to hydrogen peroxide (H_2_O_2_). CAT is an enzyme that catalyzes the elimination of H_2_O_2_ throughout the reaction catalyzed by SOD. Therefore, SOD and CAT performances alternately encourage antioxidative enzymes, which provide a preventative defense against reactive oxygen species [35,36]. Additionally, GPx is an essential protection enzyme against oxidative damage, requiring glutathione as a key factor. GPx catalyzes the oxidation of glutathione-to-glutathione disulfide at the expense of H_2_O_2_ [37]. The current report reveals that SOD concentrations were diminished in cancer-bearing animals, which may be attributed to the deterioration of antioxidant status caused by mammary carcinogenesis. It was observed that patients with breast cancer and benign breast illnesses have a lower CAT enzyme measurement. Our report also shows a reduced level of CAT in the mammary tumor control group (MTC), which may be attributed to the exploitation of antioxidant enzymes in the elimination of H_2_O_2_ via injection of LA7 cancer cells. Furthermore, GPx concentration was significantly lower in the mammary tumor control group (MTC). Antioxidant enzymes have been shown to improve breast cancer cases in response to treatment [38].

Histopathological examination detects adenocarcinomas as features of a mammary gland tumor. The adenocarcinoma of the rat mammary gland tumor is derived from the epithelium in glandular tissue, which possibly accounts for its adenomatous appearance. The nucleoli of tumor cells were often pleomorphic, and mitotic figures were abundant. Necrosis is a common characteristic of adenocarcinomas, and necrotic tumors are soft and possibly fluctuant [39]. The LA7-induced tumors in rats treated with 30 mg/kg b.w. AM and 60 mg/kg b.w. AM showed significant regression in tumor foci and dramatic improvement in overall mammary gland tissue structure, with a lower record of mitotic figures than the untreated control. However, the AM-treated groups showed tumor tubules and vascularization. A short-term AM 60 mg/kg dose significantly suppressed the whole volume of tumors per rat, but the continuation of treatment produced necrosis. The delivery of blood to newly formed tissues and tumors is a mechanism that controls growth [40]. The formation of neovascularization supplies blood to encourage angiogenesis [41]. This demonstrates that vascularization could clarify the different influences of cancer chemotherapeutic candidates on angiogenesis in the intra-tumoral area. The mechanistic actions of these agents suppress and inhibit malignant cells with support from the topoisomerase enzyme. The enzymatic effect interrupts spindle establishment in malignant cells. Alternatively, AM could be used as an anti-angiogenesis agent in breast cancer patients.

Proliferating cell nuclear antigen (PCNA) as a 36 kd highly conserved nuclear protein of DNA polymerase-delta has been identified as a valuable indicator with which to evaluate tumor cell proliferation and cancer development [42]. Variations in the genes that control the timing of occurrences in the cell cycle lead to oncogenesis. PCNA overexpression has been documented in different types of cancer, including breast cancer [43]. In our study, the expression of PCNA in mammary gland tissues during mammary carcinogenesis with and without AM treatment was examined. AM decreased the expression of this proliferative marker, which clearly revealed its antiproliferative efficacy. Our results are parallel with previous reports on AM’s antiproliferative activities against human hepatoma SK-Hep-1 cells and HT-29 colon cells [44,45]. 

The p53 tumor suppressor gene is one of the main proteins regulating cell proliferation, growth, and transformation. It is highly associated with animal and human oncogenesis and is an important controller of apoptosis [46]. Over 50% of human malignancies contain a mutant form of the p53 tumor suppressor gene [47]. Cancer-related mutant types of p53 possess a prolonged half-life that stimulates carcinogenesis and malignant aggression [48]. In the current study, we have demonstrated that treatment with AM in cancer-bearing animals significantly increased p53 protein expression, and this suggests that up-regulation of p53 has a vital role in apoptosis.

The process of apoptosis encompasses a cascade of cytoplasmic and nuclear reactions that result in morphological alterations and death of the cell [49]. This phenomenon is characterized by the generation of multimers of 180–200 base pair DNA fragments through the activation of endogenous endonucleases, causing apoptosis [50]. As with other malignancies, mammary tumorigenesis inhibits apoptosis [51]. TUNEL assay was suggested as valuable confirmation of apoptosis in breast cancer animal models [52]. Concurrently, when the rats were treated with AM, apoptosis became significant. This ability to induce apoptosis has also been shown in rats with prostate cancer [53]. 

## 4. Materials and Methods

### 4.1. Extraction and Isolation of AM from C. arborescens

The stem bark of *Cratoxylum arborescens* was collected from wild trees growing in Malaysia. A voucher specimen was deposited at the Herbarium, Department of Biology, University Putra Malaysia. The finely ground air-dried stem bark of *Cratoxylum arborescens* (1.0 kg) was extracted consecutively with hexane, chloroform, and methanol to produce 6.12, 28.18, and 40.27 g, respectively, of dark, viscous semisolid material upon solvent removal. The hexane extract was chromatographed over a vacuum column and eluted with a solvent of gradually increasing polarity to produce 26 fractions of 200 mL each. This extensive fractionation and purification of fractions 14–20 consequently yielded 160 mg (26.14%) of AM (Figure 1). The characterization and identification of AM were performed and proven using a JOEL ECA-400 spectrometer (JOEl, Tokyo, Japan) operating at 400 MHz and 100 MHz for 1H and 13C, respectively. Mass spectra data was recorded using a Shimadzu GCMS-QP5050 spectrometer (Shimadzu Co., Ltd., Kyoto, Japan). The retention time was between 7–7.5 min. One peak was observed for 15 min. The detection of the peak was performed at a wavelength of 245 nm with a Gilson absorbance detector (UV-VIS165 Gilson, Middleton, Wisconsin, USA) (Appendix A).

### 4.2. Identification of AM 

The melting point of AM was between 178–180 °C, m.p 181–182 °C. UV MeOH λmax nm (log ε): 390 (2.41), 358 (3.99), 316 (3.99), and 238 (2.65). IR νmax cm^−1^ (KBr): 3369 (OH), 2934 (CH), 1608 (C=C), 1462, and 1286. EIMS m/z (% intensity): 410 (43.06), 395 (6.14), 379 (1.61), 354 (25.77), 339 (100.00), 311 (32.57), 296 (12.89), 285 (18.90), 257 (6.46), and 162 (14.16). The 1H-NMR (500 MHz, acetone-d6): δ 13.79 (OH-1), 9.62 (OH-6), 9.52 (OH-3), 6.81 (s, 1H, and H-5), 6.38 (s, 1H, and H-4), 5.26 (t, J = 6.85 Hz, 2H, H-12, and H-17), 4.12 (d, J = 6.85 Hz, 2H, and H-11), 3.78 (OMe-7), 3.35 (d, J = 8.00 Hz, 2H, and H-16), 1.82 (s, 3H, and Me-14), 1.71 (s, 3H, and Me-19), and 1.64 (s, 6H, Me-15, and Me-20). The 13C-NMR (125 MHz, acetone-d6): δ 182.0 (C-9), 162.1 (C-4a), 160.9 (C-1), 156.6 (C-10a), 155.4 (C-6), 154.9 (C-3), 143.6 (C-7), 137.3 (C-8), 130.6 (C-18 and C-13), 123.9 (C-12), 122.6 (C-17), 111.2 (C-8a), 110.2 (C-2), 102.8 (C-9a), 101.9 (C-5), 92.3 (C-4), 62.5 (OMe-7), 26.1 (C-11), 25.1 (C-15 and C-20), 21.1 (C-16), 17.5 (C-14), and 17.1 (C-19).

### 4.3. Chemicals, Reagents, and Cell Culture 

All chemicals were obtained from Sigma Chemical Co., St. Louis, MO, USA. All reagents, solvents, and chemicals used in this study are of analytical grade. LA7 rat mammary gland tumor cells were purchased from ATCC (Manassas, VA, USA). The cells were grown in Dulbecco’s Modified Eagle’s medium (DMEM) at 37 °C in a humidified atmosphere of 5% CO_2_ supplemented with 10% fetal bovine serum (FBS), 100 μg/mL streptomycin, and 100 IU/mL penicillin. 

### 4.4. Ethical Issues

According to the ethic certificate no. (FAR/20/04/2013/MYID), the Institutional Animal Ethical Committee of the University of Malaya has sanctioned and approved this research. Pathogen-free female Sprague-Dawley rats (7–9 weeks old) were obtained from the Animal House, Faculty of Medicine, University of Malaya, Kuala Lumpur, Malaysia. They were maintained at 25 ± 3 °C with a relative humidity of 55–60 °C (a cycle of 12 h of light and 12 h of darkness) and provided with standard food pellets and tap water ad libitum.

### 4.5. Cell Preparation

When cells reached 90% confluence, the medium was replaced with fresh medium to remove dead and detached cells. The next day, the medium was removed, and the cells were washed with phosphate-buffered saline (PBS). A minimum amount of trypsin-ethylenediaminetetraacetic acid (EDTA) was added to detach the cells. The cells were obtained immediately by centrifuging them at 100 g for 10 min at 4 °C, washed twice with PBS, dispersed in PBS, and counted with a hemocytometer. Trypan blue staining was used to exclude dead cells. Eventually, cells were suspended in 300 μL PBS. All harvested cells were used within one hour of preparation.

### 4.6. Induction of Mammary Gland Tumors

After an acclimation period of one week, rats were anesthetized with an intraperitoneal injection of a mixture of ketamine-HCl (150 mg/kg body weight) and xylazine (10 mg/kg body weight). The LA7 cells (300 µL containing 6 × 10^6^ cells) were inoculated subcutaneously into the mammary fat pad (right or left flank) of each animal using a tuberculin syringe and 26-gauge needle. LA7 cell line was used as an animal model for human breast carcinoma because the cells are similar in hormone sensitivity and histopathology [16].

### 4.7. Experimental Design and Animal Treatment

A total of thirty female SD rats were used in this study. The animals were randomly divided into six groups (n = 5), where group (I) animals were kept as the normal control group, termed (NC). Group II was classified as an LA7-induced non-treated or mammary tumor control group and labeled as (MTC). Groups III–V represent the treatment groups, which include LA7-induced mammary gland tumors that were subcutaneously administered 10 days after tumor induction. They are distributed as group III treated with low-dose AM (30 mg/kg) (MT + AM-LD), group IV treated with high-dose AM (60 mg/kg) (MT + AM-HD), and group V treated with the standard drug TAM (10 mg/kg) and serving as the standard control group (MT + TAM-SC). Group VI rats were orally given AM at 60 mg/kg and served as AM control. All AM treatments were selected based on the toxicological report [54], and they were given to the animals orally twice a week for 28 days using a gastric tube. The AM extract and the standard drug Tamoxifen (TAM) were all prepared by dissolving them in Tween 20. Each animal’s body weight was recorded weekly, and blood was collected via cardiac puncture using 26-gauge needles. The blood specimens were permitted to clot prior to centrifugation at 1000× *g* for 10 min at 4 °C to gain serum. Post-experimentation, all animals were euthanized with CO_2_. Tissue samples of the mammary and liver areas were collected, rinsed twice using ice-cold 0.1 M phosphate buffer saline (PBS), dried, weighed, and preserved for biochemical and histopathological evaluations. 

### 4.8. Serum Biochemical Parameters

Blood samples were allowed to clot at room temperature and centrifuged at 1000× *g* for 10 min. The serum was separated and analyzed for tumor markers. Carcinoembryonic antigen (CEA), breast-cancer-specific marker (CA 15-3), lactate dehydrogenase (LDH), alkaline phosphatase (ALP), and alanine aminotransferase (ALT) were assessed using a chemistry analyzer (HITACHI, 902, Tokyo, Japan) using standard diagnostic kits (Roche, Rotkreuz, Switzerland).

### 4.9. Biochemical Parameters of Mammary and Liver Tissues

The biochemical activities of superoxide dismutase (SOD), catalase (CAT), glutathione peroxidase (GPx), and lipid peroxide (LPx) were conducted in rat mammary and liver tissues. Briefly, tissues were removed from experimental animals for antioxidant enzyme assessment. Tissues were rinsed thoroughly with ice-cold normal phosphate buffer saline, pH 7.2 (PBS, 0.9%), and sliced into small slices with a sharp blade. After the homogenization of the tissues using the glass homogenizer tube in cold phosphate buffer saline, mammary and liver tissues were centrifuged at 20,000× *g* for 10 min. The supernatant was mixed with phosphate buffer saline up to the last protein concentration. Influences of AM on the measurements of superoxide dismutase (SOD), catalase (CAT), glutathione peroxidase (GPx), and lipid peroxide (LPx) were assessed in the mammary and liver tissues of treated and untreated rats.

#### 4.9.1. Estimation of Superoxide Dismutase (SOD) Activity

SOD activity was measured using the Nitroblue Tetrazolium (NBT) reduction assay [55]. Consequently, 100 µL of the mammary/liver tissue supernatant supplemented a reaction mixture comprising 0.1 mM EDTA (200 µL), 0.12 mM riboflavin (50 µL), and 0.6 M phosphate buffer (pH 7.8) in a total volume of 3 mL. At 560 nm, optical density was recorded spectrophotometrically.

#### 4.9.2. Inhibition of LPx Formation

Considering the induction process of the Fe^3+/^ascorbate coupled system, Tris-HCl (40 mM), ferrous ammonium sulfate (0.16 mM), and ascorbic acid (0.06 mM) were added to 0.2 mL, 50% *w*/*v*, rat mammary and liver homogenate. After incubation for 1 h at 37 °C, the thiobarbituric reactive substances (TBARS) were evaluated [56]. An amount of 0.4 mL of the reaction mixture was added to another mixture that comprised sodium dodecyl sulfate (0.2 mL, 8%), thiobarbituric acid (1.5 mL, 0.8%), and acetic acid (1.5 mL, pH 3.5). The solution volume was increased to 4 mL using distilled water. An amount of 0.4 mL of the solution was finally preserved at 95 °C for 1 h in a water bath. A mixture of n-butanol/pyridine (15:1, *v*/*v*) was then added to the previously prepared solution after cooling, followed by shaking, centrifugation, and finally separation of the resulting organic layer. The LPx level was assessed for thiobarbituric acid formation. The intensity of the developed color was determined using a spectrophotometer at 530 nm.

#### 4.9.3. Estimation of Glutathione Peroxidase GPx Activity

Mammary and liver GPx levels were determined using the Ellman method. Concisely, 720 μL of the tissue homogenates in 200 mM Tris-HCl buffer (pH 7.2) were diluted up to 1440 μL using the same buffer. A volume of 160 μL of 5% TCA was added to the mixture and mixed thoroughly. After centrifugation at 10,000× *g* for 5 min at 4 °C, 330 μL of supernatant was allocated in a tube, and 660 μL of Ellman’s reagent (DTNB) solution was added to the mixture. Eventually, the optical density was measured at 405 nm. Protein content in each sample was determined using a bicinchoninic acid (BCA) protein assay kit (Pierce) [57].

#### 4.9.4. Estimation of CAT Activity

Mammary and liver catalase activities were evaluated and measured spectrophotometrically [58]. In short, 1.95 mL of 10 mM H_2_O_2_ in 60 mM phosphate buffer (pH 7.0) were mixed to form a 2.5 mL reaction mixture. A volume of 0.5 mL of the tissue homogenate was added to the obtained mixture, and the reaction was immediately initiated. After 3 min, each absorbance was recorded at 240 nm and compared with the absorbance of phosphate buffer (60 mM, pH 7.0), which was employed as a reference. The specific activity of CAT was determined utilizing an extinction coefficient of 0.04 mM^−1^ cm^−1^, and the results were presented as mol H_2_O_2_ consumed/(min (mg protein)).

### 4.10. Macroscopic and Microscopic Analysis

The mammary gland tumor size was measured with a caliper. Next, the tumors were immersed in 10% buffered formalin for fixation and processed for embedding in paraffin. Sections of 5 µm were cut, stained with hematoxylin-eosin, and evaluated using light microscopy (Olympus BX51, Olympus, Tokyo, Japan). Histopathological assessment was carried out with the histopathologist blinded to the treatment and based on nuclear pleomorphism, tubule formation, and mitotic score according to the Nottingham Grading System [59] and modified by Scarf and Torloni [60]. Histopathological scoring and grading were subjected to the Kruskal–Wallis test using SPSS software version 19.0 (IBM SPSS Inc., Chicago, IL, USA), and the results are illustrated in Table 3.

### 4.11. TUNEL Assay

The TUNEL assay was performed using the In Situ Apoptosis Detection Kit (Promega Inc., Madison, WI, USA) to assess the DNA fragmentation of apoptotic cells in the mammary tissues. Apoptosis was measured by joining flourescein-12-dUTP (a) at 3′-OH DNA ends utilizing the terminal recombinant deoxynucleotidyl transferase enzyme (rTdT) based on the company’s instructions. The total number of stained cells was then recorded from 5 randomly selected fields per slide under a confocal microscope (ZIESS, LSM 70, Oberkochen, Baden-Wurttemberg, Germany).

### 4.12. Immunohistochemistry Examination

As was previously performed in our laboratory [60,61], the protocol of Dakocytomation (Carpinteria, CA, USA) was followed for immunohistochemistry staining. Mammary gland sections were processed on poly-L-lysine slides for immunohistochemistry staining, and the slides were incubated in an oven at 45 °C for 30 min. The samples were then gradually rehydrated at various alcohol concentrations after deparaffinization with xylene. All target antigens were extracted using sodium citrate buffer pre-boiled in a microwave for 10 min. The tumor sections were thoroughly cleaned with newly prepared phosphate-buffered saline after the endogenous peroxidase was blocked for 5 min using the company’s provided 3% peroxidase blocking solution (PBS, pH 7.2). The various target antibodies were then incubated on the sections for 15 min within the humid chamber. The concerned antibodies, PCNA and P53, were prepared at 1:200 and 1:100, respectively. After washing, streptavidin-HRP was applied to the tissue slices, which were placed in a humid chamber for 15 min. After washing, the tumor tissues received the diaminobenzidine (DAB) substrate, chromogen, and were incubated for 5 min. The liver tissues were first cleaned with distilled water, gradually dehydrated in alcohol and xylene, and then covered to be examined under an optical microscope. The percentage of immunopositive cells was calculated using the following formula:% positive immune-stained cells = [(number of positive cells)/(total number of cells)] × 100 (64).

### 4.13. Statistical Analysis

All data were expressed as mean ± SD and analyzed using one-way ANOVA followed by post-hoc Tukey HSD multiple comparison tests. The type-1 error level was set at *p* < 0.05 for all tests. Tumor scores were subjected to the Kruskal–Wallis test. All statistical analyses were performed using SPSS software (Chicago, IL, USA), version 19.0 for Microsoft Windows^®^.

## 5. Conclusions

The present report displays for the first time that AM suppresses LA7-induced mammary carcinogenesis in rats through its antiproliferative, antioxidant, and free radical scavenging properties. Therefore, AM is a potentially effective chemotherapeutic candidate for breast cancer. This could provide new avenues for phytotherapeutic targeting and, subsequently, new alternatives for the treatment of breast cancer from natural sources.

## Figures and Tables

**Figure 1 ijms-24-10283-f001:**
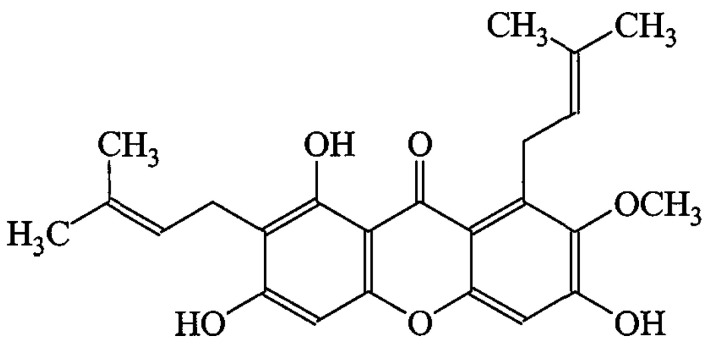
Chemical structure of α-mangostin (AM).

**Figure 2 ijms-24-10283-f002:**
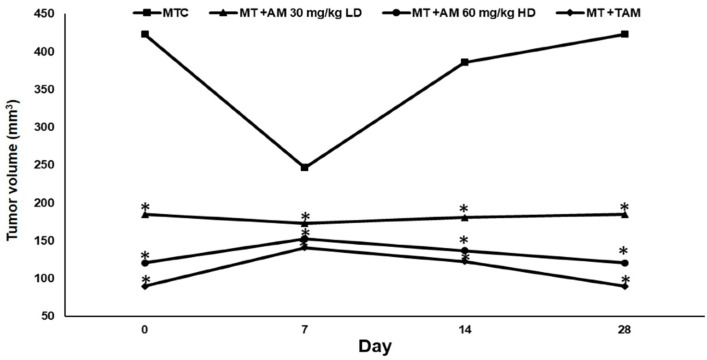
The effect of treatments on animal tumor volume (mm^3^) at 0, 7, 14, and 28 days compared to the control groups. Each value represents mean ± S.D. of given number of animals (n = 5), Values are statistically significant at * *p* < 0.05.

**Figure 3 ijms-24-10283-f003:**
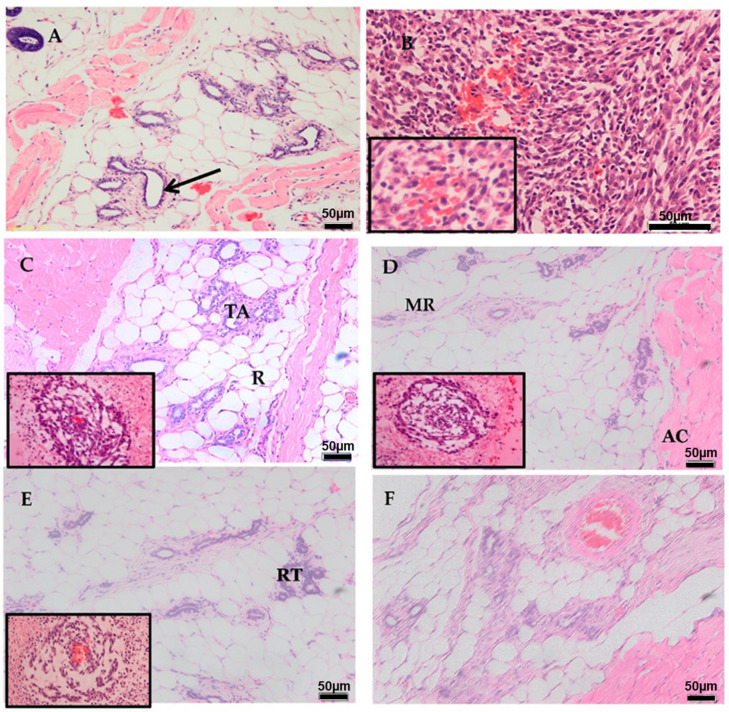
Histopathological photomicrograph of (**A**) Normal mammary gland before experimental cancer induction; note well differentiated ductal structure. Black arrow refer to normal well differentiated ductal structure. (**B**) Mammary gland after experimental tumor induction; note poorly differentiated ductal and tubular structures with variation in cellular nuclear sizes (UT). Note insert (left bottom corner) showing neovascularization disrupted tubular structure and high mitotic index. (**C**) Mammary gland treated with low dose after experimental cancer induction. Note reorganization of mammary tubular structure (MR) demarcated by muscle from tubular adenocarcinoma (AC). Note insert (left bottom corner) showing reorganization of tubular structure and neovascularization disrupted tubular structure. (**D**) Mammary gland treated with high dose after experimental cancer induction. Note reorganization of mammary tissue (R) interspersed with tubular adenocarcinoma (TA) with extensive necrosis and hyaline deposition in the connective tissue. Note insert (left bottom corner) showing organization of tubular structure and extensive necrosis of the connective tissue. (**E**) Mammary gland treated with TAM after experimental mammary tumor induction, showing ductal reorganization (RT). Note neovascularization, congestion, necrosis, and reorganization of mammary tissue interspersed with adenocarcinoma (insert). (**F**) Mammary gland treated with AM 60 mg/kg alone showing apparently normal appearance of tissue with well-differentiated ductal structure (Bar = 50 µm, H&E).

**Figure 4 ijms-24-10283-f004:**
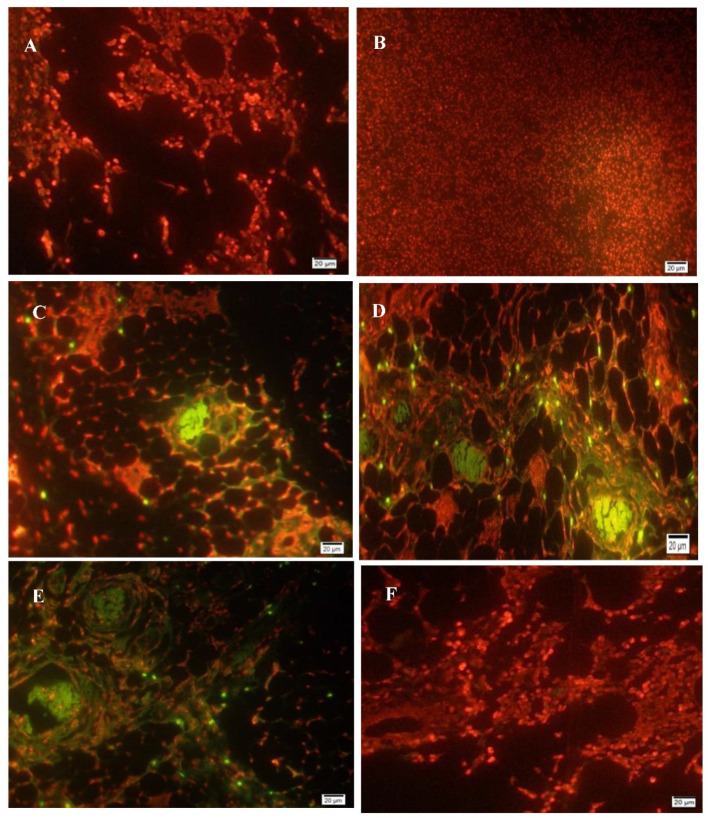
In situ TdT-mediated dUTP nick-end labeling (TUNEL assay) in breast tissue of rats. (**A**) Normal section showing absence of apoptotic cells. (**B**) TUNEL staining in (MTC group) section showing aggressive cell proliferation without apoptosis. (**C**) Cancerous section treated with 30 mg/kg body wt. AM showing numerous TUNEL-positive cells. (**D**) Cancerous section treated with 60 mg/kg body wt. AM showing frequent TUNEL-positive cells among treatment groups. (**E**) Cancerous section treated with 10 mg/kg body wt. of TAM showing the most frequent TUNEL-positive cells among treatment groups. (**F**) For an amount of 60 mg/kg AM alone no signs of apoptosis were noted on these sections (Bar = 20 µm).

**Figure 5 ijms-24-10283-f005:**
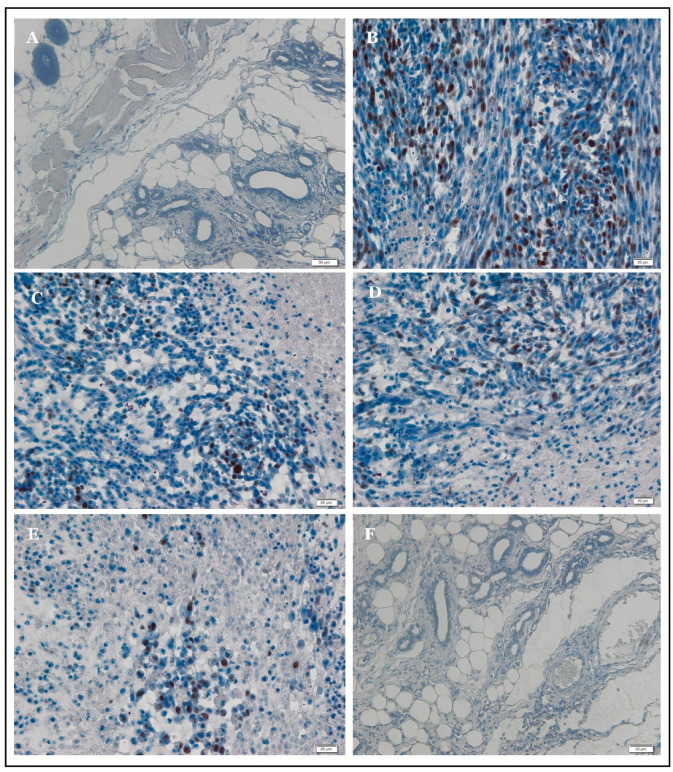
Depicts immunoexpression of PCNA in (**A**) Normal control rats (no expression), (**B**) MTC group without treatment rats (over expression) with strong diffuse intensity, (**C**) (MT + AM-LD)-treated rats (down-regulated) with strong multifocal intensity, (**D**) (MT + 60 mg/kg AM-HD) treated rats (down-regulated) strong focal intensity, (**E**) (MT + 10 mg/kg TAM) down-regulation with strong focal intensity. (**F**) AM-HD 60 mg/kg alone treated rats (expression not detectable). Brown staining indicates positive cells (20× magnifications).

**Figure 6 ijms-24-10283-f006:**
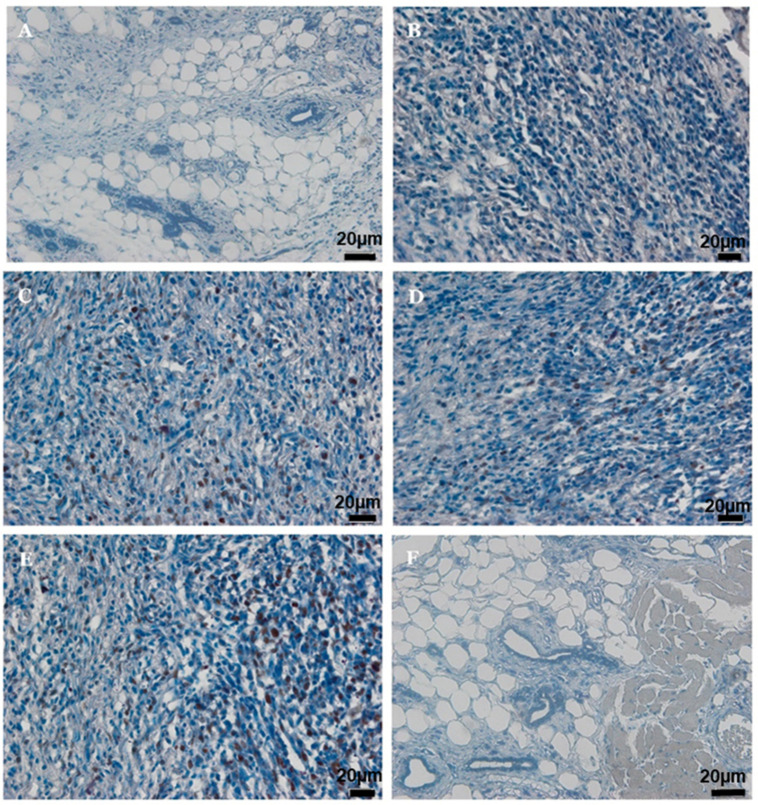
Depicts immunoexpression of p53 in (**A**) Normal control rats (weak expression), (**B**) MTC group without treatment rats (weak expression), (**C**) (MT + AM-LD)-treated rats (up-regulated) with strong multifocal intensity, (**D**) (MT + AM-HD)-treated rats (up-regulated) with strong focal intensity, (**E**) (MT + 10 mg/kg TAM) up-regulated with strong multifocal intensity, (**F**) AM 60 mg/kg alone treated rats (weak expression). Brown staining indicates positive cells (20× magnifications).

**Table 1 ijms-24-10283-t001:** Effect of treatment with AM-LD (30 mg/kg), AM-HD (60 mg/kg), and Tamoxifen (TAM) on animal body weight (g) and tumor volume (mm^3^) in experimental breast cancer in rats.

Group	Treatment Groups	Body Weight (g)	Tumor Volume (mm^3^)	Reduction of Tumor Percentage (%)
I.	NC	220.7 ± 2.4 *	0	0
II.	MTC	183.25 ± 3.31	423 ± 71.2	0
III.	MT + AM 30 mg/kg LD	199.13 ± 2.8 *	185 ± 12.8 *	56.3% *
IV.	MT + AM 60 mg/kg HD	205.25 ± 2 *	121 ± 19.3 *	71.4% *
V.	MT + TAM	209.88 ± 1.7 *	90 ± 27 *	78.7% *
VI.	AM 60 mg/kg	220.1 ± 0.5 *	0	0

Each value represents mean ± SD of given number of animals (n = 5); Values are statistically significant at * *p* < 0.05.

**Table 2 ijms-24-10283-t002:** Statues of breast biomarkers and liver biochemical parameters of animals treated with AM 30 mg/kg, AM 60 mg/kg, and TAM compared to control groups.

Group	Treatment Group	Serum Biomarkers				
		CEA (ng/mL)	CA15-3 (ng/mL)	ALT (U/L)	ALP (U/L)	LDH (U/L)
I	NC	3.1 ± 0.9 *	12.2 ± 4.0 *	45.0 ± 3.4 *	68.0 ± 7.3 *	612 ± 201.34 *
II	MTC	7.3 ± 2.3	48.0 ± 12.5	95.4 ± 16.1	453 ± 24	4562.0 ± 850
III	MT + AM 30 mg/kg LD	4.6 ± 1.2 *	32.6 ± 9.4 *	56.0 ± 14.8 *	224.0 ± 5.5 *	2163.8 ± 73.1 *
IV	MT + AM 60 mg/kg HD	3.5 ± 0.8 *	27.3 ± 4.6 *	55.5 ± 14.9 *	211.3 ± 4.1 *	2189.7 ± 114 *
V	MT + TAM	3.2 ± 1.04 *	26.2 ± 6.9 *	66.5 ± 16.5 *	199.2 ± 5.2 *	2200 ± 70 *
VI	AM 60 mg/kg	3.1 ± 1.02 *	12.3 ± 4.0 *	45.0 ± 4.2 *	68.0 ± 16.2 *	614 ± 17.15 *

Each value represents mean ± SD of given number of animals (n = 5); Values are statistically significant at * *p* < 0.05.

**Table 3 ijms-24-10283-t003:** Histopathological scoring for rats’ mammary gland tissues using Nottingham grading system.

Group	Treatment Group	Tubule	Mitotic Figure	Nuclear Pleomorphism	Grade	Total Score
I	NC	100% tubular structure	No mitotic figure	Uniform nuclear morphology	Grade I	3 ± 0.0 *
II	MTC	5% tubular formation	High mitotic figure more than 10 in high power per 10 magnification field	Moderate nuclear pleomorphism	Grade III	8 ± 0.5
III	MT + AM 30 mg/kg LD	20% tubular formation	Moderate mitotic figures	Several variation in nuclear sizes	Grade II	6 ± 0.2 *
IV	MT + AM 60 mg/kg HD	50% tubular formation	Moderate mitotic figure	Moderate nuclear pleomorphism	Grade I	5 ± 0.3 *
V	MT + TAM	30% tubular formation	Low mitotic figure	Few nuclear pleomorphisms	Grade I	4 ± 0.1 *
VI	60 mg/kg AM	95% tubular structure	No mitotic figure	Uniform nuclear morphology	Grade I	3 ± 0.1 *

Each value represents mean ± SD of given number of animals (n = 5); Values are statistically significant at * *p* < 0.05; Score 3–5 (low or grade I); Score 6–7 (intermediate or grade II); Score 8–9 (high or grade III).

**Table 4 ijms-24-10283-t004:** The effect of treatment with AM 30 mg/kg, AM 60 mg/kg, and TAM on antioxidant enzymes on the liver in experimental breast cancer in rats.

Group	Treatment Groups	SOD (U/mg Protein)	CAT (µmol/mg Ptotein)	GP_X_ (µg/mg Protein)	LPx (nmol/mg Protein)
I	NC	8.42 ± 1.12 *	62.32 ± 2.15 *	7.24 ± 1.56 *	1.21 ± 0.27 *
II	MTC	5.00 ± 1.7	39.61 ± 4.21	4.00 ± 0.23	3.13 ± 1.25
III	MT + AM 30 mg/kg LD	6.20 ± 0.47 *	56.30 ± 3.43 *	5.02 ± 0.93 *	2.12 ± 0.64 *
IV	MT + AM 60 mg/kg HD	7.24 ± 2.1 *	59.40 ± 3.23 *	5.93 ± 1.2 *	1.12 ± 0.32 *
V	MT + TAM	7.29 ± 1.23 *	61.13 ± 3.28 *	6.56 ± 0.31 *	1.18 ± 0.61 *
VI	60 mg/kg AM	8.43 ± 2.11 *	62.10 ± 2.27 *	7.22 ± 0.21 *	1.21 ± 0.11 *

Each value represents mean ± SD of given number of animals (n = 5); Values are statistically significant at * *p* < 0.05.

**Table 5 ijms-24-10283-t005:** The effect of treatment with AM 30 mg/kg, AM 60 mg/kg, and TAM on antioxidant enzymes on the breast in experimental breast cancer in rats.

Group	Treatment Groups	SOD (U/mg Protein)	CAT (µmol/mg Protein)	GP_X_ (µg/mg Protein)	LP_X_ (nmol/mg Protein)
I	NC	6.32 ± 1.91 *	57.55 ± 1.33 *	5.13 ± 1.22 *	1.81 ± 0.36 *
II	MTC	2.56 ± 0.95	23.17 ± 16.1	2.27 ± 0.06	4.19 ± 0.13
III	MT + AM 30 mg/kg LD	4.29 ± 0.08 *	48.73 ± 0.15 *	3.81 ± 0.09 *	3.02 ± 0.02 *
IV	MT + AM 60 mg/kg HD	5.24 ± 2.1 *	59.40 ± 3.23 *	4.41 ± 1.4 *	2.21 ± 0.41 *
V	MT + TAM	5.29 ± 0.08 *	56.27 ± 0.12 *	4.83 ± 0.26 *	2.02 ± 0.17 *
VI	60 mg/kg AM	6.31 ± 0.01 *	57.58 ± 1.27 *	5.13 ± 1.15 *	1.81 ± 0.25 *

Each value represents mean ± SD of given number of animals (n = 5); Values are statistically significant at * *p* < 0.05.

**Table 6 ijms-24-10283-t006:** Effect of AM on the expression of mammary PCNA and p53 proteins during mammary carcinogenesis induced by LA7 cells in rats.

Group	PCNA-Immunopositive Cells (%)	p53-Immunopositive Cells (%)
NC	0.00 ± 0.00	5.10 ± 0.96
MTC	73.76 ± 22.90 ^a^	1.50 ± 0.42 ^a^
MT + AM-LD	41.35 ± 12.67 ^a,b^	3.7 ± 1.13 ^a,b^
MT + AM-HD	32.56 ± 9.85 ^a,b^	4.20 ± 0.91 ^a,b^
MT + TAM	30.89 ± 10.12 ^a,b^	4.50 ± 1.22 ^a,b^
AM-HD	0.00 ± 0.00 ^Ns,b^	5.0 ± 1.28 ^Ns,b^

Each value represents mean ± SD of given number of animals (n = 5), significantly *p* < 0.05; Ns: not significant. Comparison ^a^: groups II, III, IV, V, and VI with group I; ^b^: groups III, IV, V, and VI with group II.

## Data Availability

Not applicable.

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
