# Peer review of "Potential Antitumor Effect of α-Mangostin against Rat Mammary Gland Tumors Induced by LA7 Cells"

_ijms, 2023, doi:10.3390/ijms241210283_

Round 1
Reviewer 1 Report (Previous Reviewer 1)
File attached

Author Response
We thank the referees for their efforts to help us improve the manuscript. The changes are highlighted in yellow in the manuscript. We responded Reviewers’ comments point by point
Reviewer 1 comments:
Page 3 – new paragraph – line 65 – it looks like instead of a number of the cited paper, the authors put detailed information about the paper – this should be in the Reference list. The same refers to line 70 (also in this paragraph).
Response: Correction made according to your comment. The detailed information about the paper was removed, and the number of cited papers was listed.
One of my previous comments was: On the other hand, in such papers, a standard is to show the plot time vs. tumor volume. I am wondering why authors do not show this in their paper. The authors responded that all correction were made according to your comments. Then where is the plot showing time vs tumor volume?
Response: Correction made according to your comment. The figure was added in the text (Figure 2, Line 108). We add here also the table showed time vs tumor volume.
The table has attached

Reviewer 2 Report (New Reviewer)
The research paper “Antitumor effect of α-mangostin against rat mammary gland tumors” by N.M. Hashim and S.N. Jayash describes in vivo evaluation of antitumor properties for α-mangostin in animal models bearing human xenografts LA-7.
Unfortunately, I can’t recommend the manuscript for publication in the IJMS due to a number of reasons:
1) The authors basically repeat their previous work ( https://www.tandfonline.com/doi/full/10.2147/DDDT.S66105 ) where they already used α-mangostin for the treatment of LA-7-bearing mice. Therefore, the scientific significance of the manuscript is questionable.
2) The MA dose required for the inhibition of tumor’s growth is ca. 6 times higher than that of TAM. Considering this fact, its not obvious if MA is a promising candidate for the breast cancer treatment.
Author Response
We thank the referees for their efforts to help us improve the manuscript. The changes are highlighted in yellow in the manuscript. We responded Reviewers’ comments point by point
Reviewer 2 comments:
English very difficult to understand/incomprehensible
Response: Correction made and to simplify and make the language easy to understand, all parts of this manuscript have been thoroughly revised.
Unfortunately, I can’t recommend the manuscript for publication in the IJMS due to a number of reasons:
1) The authors basically repeat their previous work https://www.tandfonline.com/doi/full/10.2147/DDDT.S66105 where they already used α-mangostin for the treatment of LA7-bearing mice. Therefore, the scientific significance of the manuscript is questionable.
Response: With all due respect to the reviewer's point of view, however, in our previous scientific paper published in 2014, we only touched on the ability of this promising compound to reduce the size of the cancerous tumor without addressing its mechanism of action. This manuscript describes in detail the mechanism of action of this promising compound against mammary gland tumors induced by LA 7 cells in experimental animals.
2) The AM dose required for the inhibition of tumor’s growth is ca. 6 times higher than that of TAM. Considering this fact, it is not obvious if AM is a promising candidate for the breast cancer treatment.
Response: Also, with full appreciation and respect for the reviewer's opinion, the side effects of tamoxifen used in this study compared with α-mangostin (AM), such as the development of liver cancer, increased blood clotting, retinopathy, and corneal opacity, led to the consideration of other alternative cancer treatment options. for breast cancer treatment. In addition to that, our previously published scientific paper regarding the toxicity of α-mangostin (AM) titled " α-Mangostin from Cratoxylum arborescens: An in vitro and in vivo toxicological evaluation" confirmed that it has no side effects on liver and kidney tissues up to 1000 mg/kg.
Please check this link: https://www.sciencedirect.com/science/article/pii/S1878535213003870
Round 2
Reviewer 2 Report (New Reviewer)
I would like to thank the authors for revising the manuscript
This manuscript is a resubmission of an earlier submission. The following is a list of the peer review reports and author responses from that submission.
Round 1
Reviewer 1 Report
The paper presented by Ibrahim et al. describes chemotherapeutic activity of α-mangostin against rat mammary gland tumors. While the paper is interseting, it suffers from many flaws making the read very difficult. Therefore, before publication it requires serious revision.
First of all, the authors are strongly encourage to introduce explanation of each used abbreviation (at the first time it appears in the text). For example – in Paragraph 2.2. - what does MTC stands for? – I spent some time on deciphering what MTC is. I guess the authors are using this abbreviation for mammary tumor control. But what about the others? It makes the reading of the paper much more difficult. It has to be fixed in the revised version of the manuscript. Additionally, the description of the Table 1 should indicate that the tumor volumes and body weights listed in the table are measured at 28th day of the experiment. On the other hand, in such papers a standard is to show the plot time vs. tumor volume. I am wondering why authors do not show this in their paper.
I am also wondering what idea stand behind the introduction of additional labeling of rat groups treated with 30 mg/kg and 60 mg/kg with LD and HD, respectively. The authors are asked to choose only one option.
Second, some of the western blots are not labeled (Figure 4) with the name of the detected protein. I guess the lowest one is actin but this is another thing that the reader has to guess when going through the paper.
Finally, I found several typos in the text. These are for example:
Abstract: line 27: PCNA compared to the AM–treated rats .Using the TUNEL test, there were higher numbers of
Introduction: line 41: but they suffer from itsadverse effects
Results: line 77: 423±71.2 mm3 by day 28. Meanwhile, groups treated with AM 30 mg/kg and 60 mg/kg
Reviewer 2 Report
After evaluating the manuscript "Antitumor effect of α-mangostin against rat mammary gland tumors induced by LA7 cells " I have to recommend its rejection in the current version.
Unfortunately, I want to note that the α-mangostin molecule and its antitumor properties have already been well studied (for 2022 alone, more than 15 works are already known, see for example 10.1186/s12885-022-09414-6, 10.1016/j.jff.2022.105107, 10.1158/1535-7163.MCT-20-0787, 10.3390/biomedicines10051116). Also, from the point of view of medicinal chemistry, α-mangostin is a rather metabolically active molecule that, without further hid to lead optimization, is of no interest as a real therapeutic agent. The publication of this work will rather contribute to the emergence of PAINS noise in the scientific literature.